# Exploring Markov-chain-style Evidence Using LLMs for Complex Claims

## Abstract

The remarkable success of rationale generation provokes the precise evidence discovery, which aims to identify a small subset (evidence) from the context to infer a target claim. However, existing general methods often fall short in accurately modeling evidence strength and collaborative support. This paper reformulates evidence discovery as a multi-step prompt construction process and introduces a heuristic search framework, named McsE, to explore **M**arkov-**c**hain-**s**tyle **E**vidence. Specifically, we propose a novel strength modeling perspective: Large Language Models (LLMs) can effectively serve as reward functions to estimate evidence strength when appropriately prompted. Then, we incorporate independent and collaborative reward mechanisms to systematically explore diverse potential reasoning paths, ultimately establishing the most effective prompt path as evidence. Experiments conducted on three widely-used datasets show that the proposed framework outperforms seven baselines, with distinct advantages in extracting complex evidence.

## 1 Introduction

Recently, large language models (LLMs) are steadily improving on making decisions and question-answering (Wang et al., 2019; Srivastava et al., 2022; Touvron et al., 2023; Team et al., 2024). But users still can't easily trust any given claim a model makes, since language models can hallucinate convincing nonsense (Maynez et al., 2020; Ji et al., 2023). To ensure trustworthiness and reliability, many rationalization methods emphasize leveraging evidence to generate prediction results, such as self-supported question-answering (Menick et al., 2022; Huang et al., 2024) and shortcuts discovery (Yue et al., 2024). Although high-quality evidence plays a critical role in trustworthy and explainable artificial intelligence, answering "*which parts of the context should drive model to generate a given claim?*" (evidence discovery) is still a relatively unexplored task.

There are two tasks that are related to our work: evidence retrieval (Cartright et al., 2011; Bellot et al., 2013) and evidence detection (Rinott et al., 2015). However, evidence retrieval usually focuses on identifying **whole** documents or training **task-specific** re-

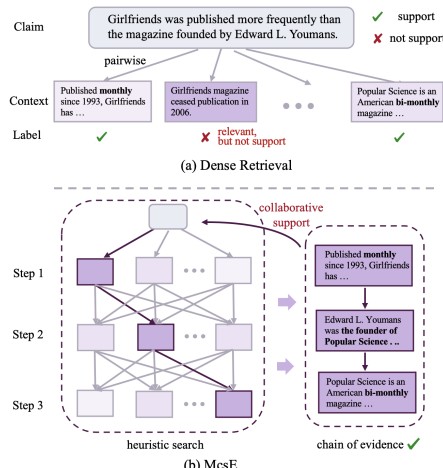

Figure 1: (a) Pairwise dense retrieval, with darker colors indicating higher relevance. (b) The proposed heuristic search framework, where darker colors represent stronger support.

trievers through supervised learning, and evidence detection's goal is to pinpoint an independent text segment which can be used **directly** to support a claim, similar to Textual Entailment (Dagan et al., 2010). In this work, we aim to construct sentence-level collaborative support. Additionally, we also highlight the difference between evidence discovery and retrieval-augmented generation (RAG): RAG retrieves external knowledge to enhance responses, while evidence discovery identifies specific segments within a pre-specified context that support deriving a target claim, emphasizing explainability.

Existing general methods often adopt off-the-shelf embedding models or LLMs to retrieve candidate text segments (Guo et al., 2022; Wang et al., 2024a; Zhu et al., 2023). Unfortunately, they have two obvious drawbacks. Firstly, **relevance does not equate to support** (Zhang et al., 2023; Wu et al., 2024). As shown in Figure 1, pairwise relevance retrieval only considers the superficial features of sentences and does not evaluate whether they truly support the claim. Secondly, evidence often doesn't appear in the form of a single sentence (Cattan et al., 2023). Previous work fails to **adequately capture inter-sentence interactions**, which limits the exploration of potential reasoning paths and leads to evidence omission. To address the above issues, we propose a heuristic search framework named McsE, to explore Markov-chain-style Evidence.

For accurately assessing evidence strength, we turn to LLMs reasoning with a carefully designed prompt. Recent studies have demonstrated that LLMs can be effectively guided by natural language prompts (Ganguli et al., 2023; Wan et al., 2023; Weller et al., 2024). Inspired by this discovery, we propose a prompt structured as "*according to...*", which **guides LLMs to ground their responses in the given context**. More importantly, in this paper, we verify that LLMs are sensitive to the strength of evidence. When guided by the *according to* prompt, LLMs tend to omit relevant information that does not align with the target claim.

To adequately capture inter-sentence interactions, we draw inspiration from the robust exploration capabilities of heuristic search algorithms and formalize evidence discovery as a Markov Decision Process. Specifically, we design a novel reward function, which **incorporates both independent and collaborative supports**, to evaluate the gain of each candidate sentence. Here, independent support refers to evaluating each candidate in a context-independent manner, while collaborative support considers the interactions among the candidates. Our framework facilitates the discovery of a more comprehensive set of evidence by leveraging multi-step evidence prompt construction.

The key contributions of this paper are:

- Formulates evidence discovery as a multi-step prompt construction process and proposes a heuristic search framework to construct Markov-chain-style evidence.
- Proposes a novel reward mechanism that explicitly explores diverse reasoning paths to construct collaborative support.
- Introduces a robust evidence strength modeling that leverages context-grounded prompting to filter out non-supporting information.
- Experiments results show that McsE can uncover complex reasoning paths, enabling more thorough evidence discovery.

## 2 RELATED WORK

### 2.1 EVIDENCE DISCOVERY IN DIFFERENT TASKS

In context-sensitive scenarios, claim attribution is crucial for both developers and users. Studies on abstractive summarization (Dou et al., 2021; Wang et al., 2022; 2024b) seek different types of guidance to support outputs, while Liu & Lapata (2019) uses a greedy algorithm to identify evidence sets. In generative QA and fact-checking, an evidence retriever is often employed to query background corpora for relevant sentences (Thorne et al., 2018; Augenstein et al., 2019; Su et al., 2021; Huang et al., 2023). Despite its importance, evidence discovery remains task-specific and often requires costly **manual annotation** (Hanselowski et al., 2019; Kotonya & Toni, 2020; Zhang et al., 2023). Addressing this, we propose a general evidence discovery framework for diverse scenarios.

### 2.2 GENERAL EVIDENCE DISCOVERY BASED ON THE INDEPENDENT SCORING PARADIGM

Traditional evidence discovery usually performs pairwise comparisons between the target claim and each candidate, which can be classified into two categories: statistical-based and embedding-based. Specially, statistical-based methods, such as BM25 or ROUGE (Robertson et al., 2009; Liu & Lapata, 2019), rank a set of candidates based on the query terms appearing in each candidate, regardless of their proximity within the context. To address this issue, embedding-based methods use contextual semantic features from pre-training. Embeddings make it possible to represent both

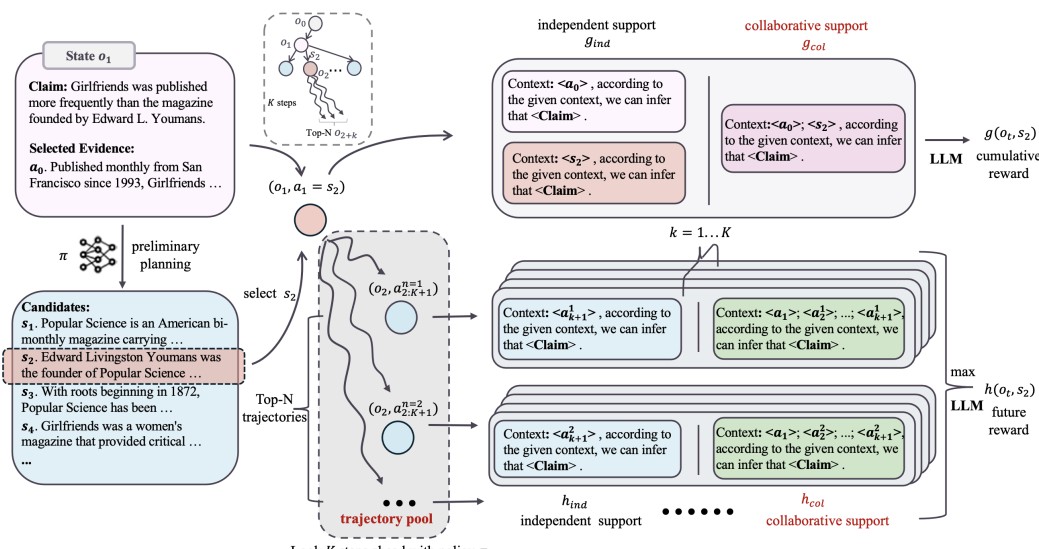

Figure 2: An illustration of our framework at step $t = 1$. It demonstrates how the reward value is calculated for selecting the candidate sentence $s_2$, where $a_0$ represents the sentence chosen at the previous step $t = 0$. Specifically, at step $t = 1$, the reward calculation for candidate sentence $s_2$ consists of two components: 1) the cumulative reward $g$ (Eq. 3); 2) the future reward $h$ (Eq. 4). McsE comprehensively considers both cumulative and future rewards to select the most valuable candidate for expanding the evidence set (Eq. 2).

candidates and claims as dense vectors in a high-dimensional semantic space and then use similarity scores for nearest-neighbor retrieval (Soleimani et al., 2020; Wang et al., 2024a). However, this independent scoring paradigm fails to capture the interactions among sentences, and ignores whether the selected evidence is sufficient.

### 2.3 LLM-BASED EVIDENCE DISCOVERY

Recently, LLM-based methods have attracted an increasing amount of attention in the information retrieval field (Sun et al., 2023a; Qin et al., 2024). For instance, Zhuang et al. (2024) attempt to use LLMs to generate the relevance between a claim and a single document. Ma et al. (2023) and Sun et al. (2023b) design *listwise* prompt for text retrieval, which insert the claim and a text list into the prompt and instruct the LLMs to output the reranked text identifiers. Although prompted LLMs have improved retrieval accuracy by enabling more nuanced matching between claims and sources (Zhu et al., 2023), we remain skeptical about whether this sequence-to-sequence generation paradigm can effectively explore the collaborative effects of evidence.

## 3 METHODS

### 3.1 TASK DESCRIPTION

We introduce several concepts which will be used throughout this paper. **Claim**: a general, concise statement that something is the case, typically query-based or aspect-based. **Context**: a set of sentences potentially relevant to the claim, usually sourced from open-source news or articles. **Evidence**: any sentence of the context that supports the claim. For the purpose of this work, we assume that we are given a concrete claim $c$ and potentially relevant context $S = \{s_0, s_1, \ldots, s_n\}$, provided either manually or by automatic methods (Roush et al., 2024; Levy et al., 2014). The task, evidence discovery, aims to automatically extract an evidence set $E = \{e_0, e_1, \ldots, e_m\}$ from the unstructured context $S$ that **support** the given claim $c$. It is worth noting that, unlike fact-checking (Thorne et al., 2018), evidence discovery assumes that the claim is partially or entirely correct based on the context.

## 3.2 Formulate Evidence Discovery as a Markov Decision Process

We first formalize the evidence discovery task as a Markov Decision Process (MDP) and then introduce a heuristic search algorithm to construct evidence prompts step by step. Referring to the classical finite Markov Decision Process (MDP), we define the four ingredients of McsE namely states, actions, transitions and rewards as follows: **State**: a state $o$ is a tuple $(c, \hat{E})$ for $c$ a claim and $\hat{E} = \{a_0, a_1, \ldots, a_k\}$ a set of sentences already selected from the context $S$. **Action**: an action $a$ is a sentence in the given context $S$. **Transition**: a transition $\mathcal{T}$ at step $t$ is a tuple $(o_t, a_t, o_{t+1})$, where $o_t = (c, \hat{E}_t)$, $o_{t+1} = (c, \hat{E}_{t+1})$ and $\hat{E}_{t+1} = \hat{E}_t \bigcup a_t$. **Reward**: the reward $\mathcal{R}$ for a transition $(o_t, a_t, o_{t+1})$ is to measure how well the claim $c$ is supported by $o_{t+1}$. Typically, we employ LLMs to generate policy $\pi(a_t|o_t) = P(a_t|o_t)$, where $a_t \in S - \hat{E}_t$. The policy $\pi$ tends to select candidates related to the preceding context, which helps maintain consistency in reasoning. In practice, we also introduce a length penalty to balance candidates of different lengths. Based on the LLM policy $\pi$, the value of transition $(o_t, a_t, o_{t+1})$ is given by a Q-function:

$$Q_\pi(o_t, a_t) = \mathbb{E}_\pi \left[ \sum_{k=0}^{K} \gamma^k \mathcal{R}(a_{t+k}, o_{t+k}) \right]. \tag{1}$$

## 3.3 Optimal Q-value Estimation Combining Independent and Collaborative Supports

Refer to the important heuristic search algorithm A* (Hart et al., 1968), McsE uses a value function $f$ to approximate the real $Q$-function, aming to overcome the vast and complex search space. Unlike previous approaches that rely on supervised learning to fit the Q-function Zelikman et al. (2022), we design an unsupervised value function to evaluate the reward of taking action $a_t$ at the state $o_t$. Specifically, $f$ is defined as:

$$f(o_t, a_t) = g(o_t, a_t) + \gamma h(o_t, a_t), \tag{2}$$

where $g(o_t, a_t)$ represents the cumulative reward of state $o_t$ after taking action $a_t$, and $h(o_t, a_t)$ denotes a heuristic function for estimating the expected future reward of taking action $a_t$. Unlike previous works that only consider independent relevance scores (Soleimani et al., 2020; Wang et al., 2024a), we introduce collaborative support in $g(\cdot)$ and $h(\cdot)$ to explore the interactions among sentences. Besides, $\gamma$ is a discount factor used to balance the importance of $g(\cdot)$ and $h(\cdot)$.

**Cumulative Reward.** As shown in Equation equation 3, the cumulative reward $g(o_t, a_t)$ consists of two parts: $g_{ind}(o_t, a_t)$, assessing the independent contribution of each $a_{t'}$ to $c$ in a context-independent manner; $g_{col}(o_t, a_t)$, evaluating the reasoning path $a_{0:t}$ by concatenating $a_t$ with $a_{0:t-1}$, which explores the 'chemical reaction' between $a_t$ and the selected evidence. We use $\lambda$ to balance $g_{ind}(\cdot)$ and $g_{col}(\cdot)$. The $score(\cdot)$ function is used to evaluate how strongly the evidence supports the claim, please refer to Section 4.4.1 for more details.

$$g(o_t, a_t) = g_{ind}(o_t, a_t) + \lambda g_{col}(o_t, a_t)$$
$$s.t. \quad \begin{cases} g_{ind}(o_t, a_t) = \frac{1}{t} \sum_{t'=0}^{t} score(c, a_{t'}) \\ g_{col}(o_t, a_t) = score(c, a_{0:t}) \end{cases} \tag{3}$$

**Future Reward.** A heuristic function $h(o_t, a_t)$, similar to $g(o_t, a_t)$, is introduced to estimate the potential future benefit of taking action $a_t$. As shown in Figure 2, starting from the state-action pair $(o_t, a_t)$, we perform rollout with policy $\pi$ to form a trajectory pool, representing different reasoning paths. In practice, we usually select the top-N trajectories with the highest probabilities to reduce computational costs. Then, the highest future reward is regarded as the potential value of taking action $a_t$. The purpose of $h(o_t, a_t)$ is to provide guidance on which unselected context sentences might, together with $(o_t, a_t)$, form a evidence path that strongly supports the given claim $c$. In Equation equation 4, $K$ is a hyperparameter used to determine how many steps to look ahead, and $\delta$ is a balancing factor. By introducing this function, our search framework can prioritize exploring states that appear to be more likely to derive the target claim, thus reducing the overall search time

and making the search process more efficient.

$$h(o_t, a_t) = \max_{\substack{\mathcal{T} \sim \pi \\ a_{t+k} \in \mathcal{T}}} \sum_{k=1}^{K} \gamma^{k-1} [h_{ind}(o_{t+k}, a_{t+k}) + \delta h_{col}(o_{t+k}, a_{t+k})] \tag{4}$$

$$s.t. \quad \begin{cases} h_{ind}(o_t, a_t) = score(c, a_{t+k}) \\ h_{col}(o_t, a_t) = score(c, a_{t:t+k}) \end{cases}$$

**Exploring Markov-chain-style Support.** Algorithm 1 in **Appendix** C provides an overview of McsE. Specifically, McsE uses a greedy strategy to determine how to expand evidence set. At each iteration of the main loop, we construct a prompt prefix based on the already selected evidence, and compute an $f$-value for each candidate $s_i$, which estimates the total reward obtained by expanding $s_i$. Then, the candidate with the highest $f$-value is selected to update state $o_t$. The algorithm continues until a specified number of sentences are selected.

### 3.4 Robust Evidence Strength Modeling via Grounding LLMs in Context

In this section, we discuss how to model evidence strength. As we all know, the generation of common LLMs is auto-regressive, where the prediction of the next token depends on the previous context. Similarly, humans logically derive conclusions by organizing and integrating known information, a process known as deductive reasoning. Therefore, this work assumes that LLMs are excellent deducers, capable of accurately assessing evidence strength: the more supportive the prompt is, the higher the probability that LLMs will generate the target claim. However, considering that LLMs may tend to produce outputs that deviate from the input, known as hallucination or inconsistency, we introduce a *according-to* prompt to ground LLMs' output in context $\hat{S}$. Figure 2 shows the proposed prompt. Finally, we force LLMs to decode the target claim $c$ and compute its log probability as score:

> **Model Input**
>
> **Claim:** Girlfriends was published more frequently than the magazine founded by Edward L. Youmans.
> **Context:** (1) Popular Science is an American bi-monthly magazine carrying popular … (2) Popular Science has won over 58 awards, including the American Society of …
>
> ---
>
> **Grounding via *according-to* Prompting:**
>
> Context: **<Context>** , according to the given context, we can infer that **<Claim>** .

Figure 3: Prompting LLMs to ground in context.

$$score(c, \hat{S}) = \sum_{i=1}^{|c|} \log P(c_i | c_{<i}, \text{prompt}(\hat{S})). \tag{5}$$

## 4 Experiments

### 4.1 Settings

**Datasets.** We conduct experiments on three common benchmarks, including HoVer (Jiang et al., 2020), PubMedQA (Jin et al., 2019), and CovidET (Zhan et al., 2022). Among these datasets, HoVer is a multi-hop dataset with manually annotated evidence, ensuring the claims are abstract in nature. However, since HoVer was originally created for fact-checking purposes, some of the claims may not be correct. Therefore, we selected instances labeled as true for testing phase. Besides, PubMedQA is a generative question-answer dataset in the biomedical field, while CovidET is an abstract summarization dataset in the COVID-19 domain. These two datasets challenge the extraction methods' generalizability across different domains. However, since both datasets lack evidence annotations, we use GPT-4 to annotate 200 random instances from each dataset, followed by manual correction for evaluation purposes. The prompt for annotation is shown in Apendix D.

**Implementation Details.** We use Gemma-2b-it[1] to generate the policy $\pi$ and model support, its advantages lie in its lightweight design and strong inference performance. The implementation of our framework based on transformers library[2]. Specifically, the hyperparameters $\gamma$, $\delta$, and $\lambda$ are set to 0.9, 1, and 1, respectively. When exploring potential reasoning paths to obtain the maximum

---

[1]https://huggingface.co/google/gemma-2b-it
[2]https://github.com/huggingface/transformers

future reward, we look ahead $K = 4$ steps and calculate the $N = 10$ paths with the highest probabilities. Following previous works (Zhang et al., 2023), we use Precision, Recall and F1 score as the evaluation metrics for evidence discovery.

## 4.2 BASELINES

we select several representative general methods as baselines: **ROUGE** (Chin-Yew, 2004): count the number of overlapping units between the candidates and the given claim. **BM25** (Robertson et al., 2009): rank candidates based on the claim term occurrence and rarity across the whole context. **MPNet** (Song et al., 2020):use the all-mpnet-v2-base version[3] to calculate the similarity between the sentence embeddings of each candidate and the given claim. **GTE** (Li et al., 2023): a general text embedding model trained with multi-stage contrastive learning, we use GTE-large[4] to calculate the candidate-claim similarity. **Gemma-Retriever**: concatenate all candidate sentences as input and prompts Gemma-7b-it[5] to directly generate the top-k most relevant sentences. **Gemma-Reranker**: concatenate all sentences that pass the initial filter by the GTE-large model as input and prompts Gemma-7b-it to rerank these candidates. **RankGPT** (Sun et al., 2023b): similar to gemma-retriever, a listwise prompting-based approach using GPT-3.5-turbo-1106.

## 4.3 MAIN RESULTS

Table 1 compares McsE's performance with state-of-the-art baselines under the topk-3 and topk-5 settings. We highlight three key observations: 1). McsE consistently outperforms various baselines across different tasks. In contrast, none of the baseline approaches consistently perform well across all three datasets. 2). Statistical-based methods exhibit the worst performance when dealing with relatively abstract claims. LLM-based methods, such as Gemma-Retriever and RankGPT, show no significant advantage over embedding-based methods, and their performance on the PubMedQA dataset is even lower than embedding-based approaches. This finding indicates that current LLM-based methods also lack the capability to explore the collaborative effects among evidence. 3). Our McsE, implemented with Gemma-2b-it, significantly outperforms both the Gemma-Retriever and Gemma-Reranker based on Gemma-7b-it, acheiveing up to 3.8%-6.7% higher F1 score than Gemma-Retriever and 3.8%-6.7% higher F1 score 1.2%-5.1% than Gemma-Reranker.

Table 1: Results on HoVer, PubMedQA and CovidET Datasets.

| Method Type | Method | HoVer | | | PubMedQA | | | CovidET | | |
|---|---|---|---|---|---|---|---|---|---|---|
| | | P | R | F1 | P | R | F1 | P | R | F1 |
| **Top-3** | | | | | | | | | | |
| Statistical-based | ROUGE | 57.3 | 52.4 | 54.8 | 39.0 | 45.1 | 41.7 | 44.3 | 41.3 | 42.8 |
| | BM25 | 58.0 | 53.1 | 55.4 | 40.0 | 46.7 | 43.1 | 48.7 | 45.3 | 46.9 |
| Embedding-based | MPNet-base | 58.7 | 53.7 | 56.1 | 44.7 | 52.1 | 48.1 | 53.7 | 50.0 | 51.8 |
| | GTE-large | 60.0 | 54.9 | 57.3 | 46.0 | 53.7 | 49.6 | 55.7 | 51.9 | 53.7 |
| LLM-based | Gemma-Retriever | 59.7 | 54.6 | 57.0 | 41.6 | 48.3 | 44.7 | 55.3 | 51.5 | 53.4 |
| | Gemma-Reranker | 61.0 | 55.8 | 58.3 | 45.3 | 52.8 | 48.8 | 56.7 | 52.8 | 54.7 |
| | RankGPT | 62.2 | 55.2 | 58.5 | 43.3 | 50.6 | 46.7 | 55.9 | 51.9 | 53.8 |
| | McsE | **64.7** | **59.2** | **61.8** | **48.0** | **55.6** | **51.4** | **60.7** | **56.6** | **58.5** |
| **Top-5** | | | | | | | | | | |
| Statistical-based | ROUGE | 47.0 | 69.8 | 56.2 | 34.0 | 66.5 | 45.2 | 37.6 | 58.4 | 45.7 |
| | BM25 | 48.9 | 72.6 | 58.4 | 33.4 | 65.0 | 44.1 | 38.4 | 59.6 | 46.7 |
| Embedding-based | MPNet-base | 49.5 | 73.5 | 59.1 | 36.0 | 68.8 | 47.3 | 43.8 | 68.0 | 53.3 |
| | GTE-large | 50.5 | 75.0 | 60.3 | 36.7 | 70.1 | 48.1 | 43.6 | 67.7 | 53.0 |
| LLM-based | Gemma-Retriever | 52.2 | 72.9 | 60.8 | 34.9 | 66.2 | 45.7 | 43.1 | 63.5 | 51.4 |
| | Gemma-Reranker | 51.0 | 75.7 | 60.9 | 37.1 | 70.8 | 48.7 | 43.7 | 67.9 | 53.2 |
| | RankGPT | 53.6 | 79.6 | 64.0 | 35.9 | 68.1 | 47.0 | 44.1 | 65.2 | 52.6 |
| | McsE | **55.2** | **82.0** | **66.0** | **38.0** | **73.5** | **49.9** | **45.4** | **70.5** | **55.2** |

---

[3]https://huggingface.co/sentence-transformers/all-mpnet-base-v2

[4]https://huggingface.co/thenlper/gte-large

[5]https://huggingface.co/google/gemma-7b-it

### 4.4 ANALYSIS

#### 4.4.1 ARE LLMS SENSITIVE TO EVIDENCE STRENGTH?

Previous works have demonstrated that LLMs can be prompted to calculate the relevance between two sentences (Qin et al., 2024). However, these scoring methods often lack a point of reference, making it difficult to evaluate the variations in evidence strength. In this section, we verify that the output probability given by a LLM with *according to* prompt can serve as an effective metric. As shown in Table 2, we categorize the context into the following cases based on the support strength it provides for the claim: 1) not related. Randomly select $m$ sentences from contexts unrelated to the given claim as input; 2) not relevant. Randomly select $m$ non-evidence sentences from the context corresponding to the given claim; 3) sufficient. Concatenate all sentences in the golden evidence set as input; 4) -w/o $m$ sentences. Randomly remove $m$ sentences from the set of

Table 2: Modeling the strength of evidence. We report the average log probability (token-level) on the 4-hop HoVer development set for prompt-based methods and the average cosine similarity for embedding-based methods.

| input | *according to* -w/o | *according to* -w | GTE |
|---|---|---|---|
| not related | -4.36 | -4.47 | 71.96 |
| not relevant | -4.22 | -4.19 | 86.34 |
| sufficient | -3.23 | -2.95 | 85.45 |
| -w/o 1 sentence | -3.50 | -3.27 | 86.13 |
| -w/o 2 sentences | -3.86 | -3.72 | 85.00 |
| -w/o 3 sentences | -4.03 | -3.98 | 84.21 |
| -w/o 4 sentences[6] | -4.31 | -4.41 | / |
| -w/ not related | -3.27 | -2.99 | 85.61 |
| -w/ not relevant | -3.11 | -2.97 | 89.36 |

golden evidence, and concatenate the remaining sentences as input; 5) -w/ not related. Add not related sentences to the set of golden evidence; 6) -w/ not relevant. Add irrelevant sentences to the set of golden evidence.

Based on the results shown in Table 2, we have the following findings: 1) Without introducing additional input noise, LLMs can accurately perceive evidence strength, regardless of whether the *according to* prompt is used. However, after using the *according to* prompt, this perception becomes more sensitive and exhibits greater fluctuations; 2) The *according-to* prompt helps LLMs recognize and filter out related but irrelevant noise while enhancing their ability to resist relevant noise; 3) The feedback from LLMs using a *according to* prompt aligns with human performance across different context, making it an ideal reward function; 4) In contrast to prompt-based methods, GTE struggle to omit noise that is relevant to the given claim.

#### 4.4.2 HOW DOES THE SIZE OF THE TRAJECTORY POOL AFFECT PERFORMANCE?

McsE uses a rollout policy $\pi$ for expansion. A larger trajectory pool represents more candidate paths but increases inference cost. In Figure 4, we compare the performance of our McsE across different sizes of the trajectory pools using the 2-hop, 3-hop and 4-hop HoVer datasets. We highlight two key observations: 1). In the early part, the performance of evidence discovery improves as the number of candidate reasoning paths increases. 2). The more complex the evidence, the slower its corresponding curve converges.

#### 4.4.3 IS MCSE A GENERAL-PURPOSE EVIDENCE DISCOVERY FRAMEWORK?

Table 1 shows that McsE outperforms mainstream embedding-based and LLM-based methods across different tasks and domains. In this section, we further validate that McsE is capable of extracting evidence of varying complexity. We categorize the HoVer dataset

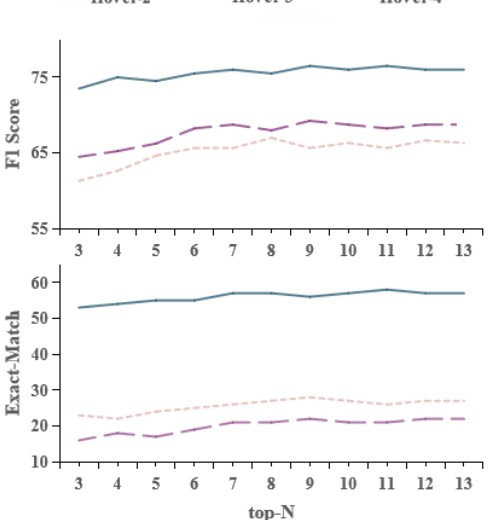

Figure 4: Performance comparisons on the HoVer dataset under different size of the trajectory pools.

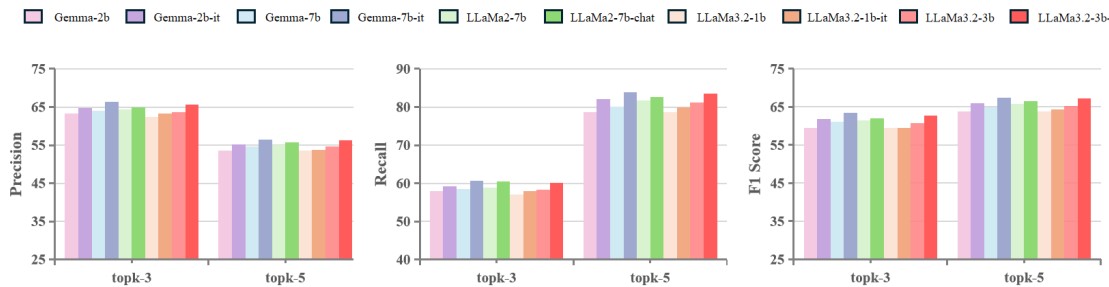

Figure 5: Performance comparisons on the HoVer dataset under different base models.

based on the number of evidence, and then randomly select 200 examples from each category for testing. We also conduct experiment on the 1-hop FEVER dataset (Thorne et al., 2018). In addition to the F1 score, we report the Exact-Match (EM) score to assess the completeness of evidence extraction. Our method shows significant improvement in extracting complex evidence, with the extent of improvement growing as the number of hops increases. Additionally, in the 1-hop scenario, McsE can achieve acceptable performance relying solely on the independent reward.

Table 3: Performance on 1/2/3/4-hop data.

| Method Type | Method | FEVER-1 | | HoVer-2 | | HoVer-3 | | HoVer-4 | |
|---|---|---|---|---|---|---|---|---|---|
| | | F1 | EM | F1 | EM | F1 | EM | F1 | EM |
| Statistical-based | ROUGE | 45.0 | 45.0 | 63.0 | 41.0 | 55.7 | 16.5 | 59.5 | 10.0 |
| | BM25 | 51.0 | 51.0 | 68.5 | 47.5 | 59.3 | 17.0 | 59.0 | 10.0 |
| Embedding-based | MPNet-base | 52.5 | 52.5 | 69.0 | 46.5 | 61.3 | 16.0 | 59.3 | 10.5 |
| | GTE-large | 50.0 | 50.0 | 73.0 | 53.0 | 59.0 | 16.5 | 59.8 | 13.0 |
| LLM-based | Gemma-Retriever | 59.5 | 59.5 | 65.0 | 43.0 | 55.3 | 14.5 | 56.0 | 9.5 |
| | Gemma-Reranker | 62.0 | 62.0 | 71.0 | 50.5 | 63.0 | 19.0 | 53.0 | 11.0 |
| | RankGPT | **70.0** | **70.0** | 68.5 | 46.5 | 61.7 | 18.5 | 63.0 | 14.0 |
| | McsE | 61.0 | 61.0 | **76.0** | **57.5** | **66.3** | **26.0** | **68.8** | **21.5** |

### 4.4.4 BASE MODEL GENERALIZABILITY

In this section, we first discuss the impact of model size and instruction fine-tuning on the performance of the proposed framework. The experimental results on the HoVer dataset are shown in Figure 5. Specifically, we compared the performance of Gemma-2b, Gemma-2b-it, Gemma-7b, and Gemma-7b-it under the top-3 and top-5 settings. We found that instruction fine-tuning yields a more significant improvement than merely increasing the model size. This is likely because instruction fine-tuning improves the model's capability to effectively follow and execute prompts. Considering the value function's result on the same test instance could be similar since both models are trained on similar lexical, we further evaluate our framework with the LLaMA series of similar size to show the generalizability: McsE's performance still remains stable.

### 4.4.5 WHAT ADVANTAGES DOSE MCSE BRING WITH RESPECT TO COT?

Chain-of-Thought has been explored as a simple and broadly applicable method for enhancing complex reasoning in language models. To compare with CoT, we construct intermediate reasoning steps based on the golden evidence to guide the language model's reasoning. Considering the creativity of the generation, we use a natural language inference model TRUE[7] and GPT-3.5-turbo-1106 to automatically assess the consistency between the generated evidence and the given claim. As shown in Table 4, McsE demonstrates a clear performance advantage, which we attribute to the heuristic

---

[6]For embedding-based methods, we cannot calculate the similarity between an empty string and the target claim.

[7]https://huggingface.co/google/t5_xxl_true_nli_mixture

search framework enhancing the reorganization and the integration of known information. Furthermore, CoT itself is prone to hallucinations and is not directly applicable for rationale generation methods.

Table 4: Compared to CoT's performance. We use in-context learning to guide different foundational models to generate explanation as evidence.

| Metric | few-shot CoT | | | McsE |
|---|---|---|---|---|
| | Gemma-2b-it | Gemma-7b-it | GPT-3.5-turbo | |
| TRUE | 33.0 | 43.5 | 47.0 | **57.0** |
| GPT-3.5-turbo | 58.0 | 70.5 | 69.5 | **82.0** |

### 4.4.6 ABLATION ANALYSIS

As shown in Table 5, removing the *according to* prompt results in the worst performance, indicating the necessity of precise evidence strength evaluation. Removing the independent rewards ($g_{ind}$ and $h_{ind}$) achieves superior performance on EM metric over removing the collaborative rewards ($g_{col}$ and $h_{col}$), demonstrating that the collaborative rewards are crucial for exploring potential reasoning paths and ensuring the completeness of complex evidence. Besides, the future reward $h(\cdot)$ is also important for extracting complex evidence. Finally, planning reasoning paths with policy $\pi$ performs better than random planning.

Table 5: Results of ablation study, w/o $\star$ indicates that only the component $\star$ is removed from McsE.

| | FEVER-1 | | HoVer-2 | | Hover-3 | | Hover-4 | |
|---|---|---|---|---|---|---|---|---|
| | F1 | EM | F1 | EM | F1 | EM | F1 | EM |
| McsE | 61.0 | 61.0 | 76.0 | 57.5 | 66.3 | 26.0 | 68.8 | 21.5 |
| -w/o *according to* | 54.0 | 54.0 | 72.0 | 53.0 | 59.0 | 20.0 | 65.3 | 18.5 |
| -w/o *collaborative* | 61.0 | 61.0 | 73.5 | 51.5 | 60.3 | 19.5 | 63.0 | 17.5 |
| -w/o *independent* | 61.0 | 61.0 | 75.5 | 55.5 | 63.7 | 24.5 | 67.0 | 22.0 |
| -w/o $h(\cdot)$ | 61.0 | 61.0 | 74.5 | 52.0 | 62.0 | 21.0 | 65.3 | 18.0 |
| -w/o $\pi$ | 61.0 | 61.0 | 76.0 | 54.0 | 65.7 | 24.0 | 68.3 | 20.5 |

## 5 CONCLUSION

In this paper, we examine a variety of evidence discovery techniques, demonstrating that they are sensitive to relevant information that does not support the target claim and struggle to capture the collaborative effect of evidence. Therefore, we propose McsE, a framework that (1) leverages context-grounded LLMs for robust evidence strength modeling, and (2) heuristically constructs evidence chains through a Markov decision process by combining independent and collaborative support mechanisms. Experimental results show that our framework surpasses widely-used baselines in extracting complex evidence.

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

## A  COMPLEXITY ANALYSIS

While it is true that our framework incurs additional computational cost compared to traditional retrieval-based methods like BM25 or embedding-based dense retrieval, this trade-off is intentional and justified by the significant improvements in evidence discovery quality, particularly for complex claims requiring multi-step reasoning. Moreover, the computational overhead is a worthwhile investment for applications where precision and explainability are paramount, such as fact-checking, scientific claim validation, or legal reasoning. The novelty of McsE lies in its ability to prioritize the overall quality of evidence, offering a paradigm shift from traditional retrieval to a reasoning-guided discovery process.

Table 6: Performance comparisons on the HoVer dataset under different size of the trajectory length.

| $K$ | F1 Score (Hover-2/3/4) | Exact Match (Hover-2/3/4) |
|---|---|---|
| 0 | 74.5 / 62.0 / 65.3 | 52.0 / 21.0 / 18.0 |
| 1 | 76.5 / 64.3 / 66.8 | 53.5 / 23.5 / 20.0 |
| 2 | 76.0 / 65.9 / 67.5 | 56.0 / 25.5 / 21.0 |
| 3 | 75.5 / 65.1 / 68.3 | 58.0 / 26.5 / 21.0 |
| 4 | 76.0 / 66.3 / 68.8 | 57.5 / 26.0 / 21.5 |
| 5 | 76.5 / 66.1 / 68.4 | 57.0 / 27.0 / 21.0 |

Here, we provide a formal explanation of the time complexity. At each iteration step, we use the beam search algorithm to compute the $T$ most probable paths (trajectory pool) for each candidate. Assuming the trajectory length is $K$ and the number of candidates is $N$, the time complexity per step is $O(N^2 KT)$. The algorithm's execution speed depends on the two hyperparameters, $K$ and $T$. In Figure 4, we have demonstrated the impact of trajectory pool size on McsE performance. Here, we further analyzed the impact of the hyperparameter $K$.

As shown in 6, When $K > 0$, McsE performs better. Based on the analysis of $K$ and $T$, we argue that McsE demonstrates a valuable trade-off (significant performance improvements with smaller $K$ and $T$), particularly for applications prioritizing precision and explainability, such as fact-checking, scientific claim validation, or legal reasoning.

## B COMPARISONS WITH TASK-SPECIFIC RATIONALE EXTRACTION METHODS

While our experiments focused on comparing McsE against widely-used general-purpose baselines, we acknowledge that task-specific approaches could provide additional context. To address this concern, we conducted experiments on the MultiRC dataset using identical experimental settings to ensure a fair comparison.

Table 7: Performance comparisons with task-specific approaches on the MultiRC dataset.

| Method | Top-1 (P/R/F) | Top-2 (P/R/F) |
|---|---|---|
| BM25 | 70.7 / 30.3 / 42.5 | 57.5 / 49.4 / 53.2 |
| ROUGE | 67.8 / 29.1 / 40.8 | 53.4 / 45.9 / 49.4 |
| MPNet-base | 71.9 / 30.9 / 43.2 | 58.0 / 49.8 / 53.6 |
| GTE-large | 72.3 / 31.0 / 43.4 | 59.4 / 51.0 / 54.9 |
| Gemma-Retriever | 71.2 / 29.4 / 41.6 | 61.7 / 47.9 / 53.9 |
| Gemma-Rerank | 73.6 / 30.3 / 42.9 | 63.0 / 48.3 / 54.7 |
| RankGPT | 77.1 / 31.7 / 44.9 | 66.2 / 56.9 / 61.2 |
| McsE (Ours) | 76.6 / 32.9 / 45.9 | 67.0 / 57.6 / 61.9 |
| Pipeline S | 66.7 / 30.2 / 41.6 | - / - / - |
| Faithful-Rationale U | 34.6 / 15.5 / 21.4 | 44.4 / 19.9 / 27.5 |
| Faithful-Rationale S | 74.3 / 33.5 / 46.1 | 65.8 / 42.3 / 51.4 |

Results for all task-specific methods, Pipeline S DeYoung et al. (2020) and Faithful-Rationale Glockner et al. (2020), are directly reproduced from their original publications. As shown in the table below, supervised methods demonstrate only marginal advantages over general-purpose approaches in the Top-1 setting, whereas McsE attains optimal performance under the Top-2 retrieval setting. These findings strongly underscore the potential of our LLM-based exploration.

## C OVERVIEW OF MCSE

---

**Algorithm 1** Overveiw of McsE.

---

**Require:**
    Claim $c$; the set of context sentences, $S$;
    LLM policy $\pi$; the maximum evidence size, $max\_step$.
**Ensure:**
    Evidence $\hat{E}$.
1: Initialize $\hat{E}_0 \leftarrow \emptyset$; $o_0 \leftarrow (c, \hat{E}_0)$; $t \leftarrow 0$.
2: **while** $t \leq max\_step$ **do**
3:     $f_{values} \leftarrow dict()$
4:     **for** $s_i$ in $\pi(\cdot|\hat{E}_t, S)$ **do**
5:         $g(o_t, s_i) \leftarrow g_{ind}(o_t, s_i) + \lambda g_{col}(o_t, s_i)$
6:         $\hat{a}_{t+1:t+K} \leftarrow \underset{\mathcal{T} \sim \pi(\cdot|\hat{E}_t \bigcup s_i, c)}{\arg\max} \sum_{k=1}^{K} \gamma^{k-1}(h_{ind}(o_{t+k}, \mathcal{T}_k) + \delta h_{col}(o_{t+k}, \mathcal{T}_k))$
7:         $h(o_t, s_i) \leftarrow \sum_{k=1}^{K} \gamma^{k-1}(h_{ind}(o_{t+k}, \hat{a}_{t+k}) + \delta h_{col}(o_{t+k}, \hat{a}_{t+k}))$
8:         $f_{values}[s_i] \leftarrow g(o_t, s_i) + \gamma h(o_t, s_i)$
9:     **end for**
10:    update $a_t \leftarrow \underset{s_i}{\arg\max} f_{values}[s_i]$; $\hat{E}_{t+1} \leftarrow \hat{E}_t \bigcup a_t$; $o_{t+1} \leftarrow (c, \hat{E}_{t+1})$; $t \leftarrow t + 1$
11: **end while**
12: **return** $\hat{E}_t$;

---

## D    PROMPT FOR ANNOTATION

```
f'''
A list of sentences is shown below. Each sentence has a number next to it.
A claim is also provided.
Respond with the numbers of the sentences you should consult to support the claim.
Example format:
Sentence 1:
<sentece 1>
Sentence 2:
<sentence 2>
…
Sentence N:
<sentence N>
Claim: <claim>
Answer:
Sent: 9
Sent: 3
Sent: 7
Let's try this now:
{input sentences}
Claim: {input claim}
Answer:
'''
```

Figure 6: The listwise prompt for evidence annotation.

