# OpenReview forum: "Exploring Markov-chain-style Evidence Using LLMs for Complex Claims"
_ICLR.cc/2026/Conference — Submitted to ICLR 2026_

### Official Review · Reviewer_xZfh · 2025-10-31

**Soundness:** 2
**Presentation:** 2
**Contribution:** 3
**Rating:** 2
**Confidence:** 4

**Summary:**

This paper addresses the problem of evidence discovery for claim verification—a vital task for explainable and trustworthy AI. The authors critique existing methods, which largely focus on simple relevance detection and often ignore the interdependence or collaborative nature of evidence. They propose McsE, a heuristic search framework that models evidence discovery as a Markov Decision Process, leveraging LLMs as reward functions to score the “strength” of evidence. Key innovations include the use of context-grounded prompts and the explicit modeling of both independent and collaborative evidence support. The framework is evaluated on multiple datasets (HoVer, PubMedQA, CovidET), demonstrating superior performance over both statistical, embedding-based, and LLM-based baselines.

**Strengths:**

1. This paper is clearly written.
2. Evidence discovery is formulated as a Markov Decision Process, allowing a systematic exploration of reasoning paths rather than independent selection.

**Weaknesses:**

**My main concern is that the build retrieval MDP may be equivalent, even only a subset of the active retrieval or active RAG methods**

There is a wide collection of active retrieval and generation methods, including [1], the famous self-RAG, [2], the famous IterGen. These methods build the new retrieval query by synthesizing the unsolved part of the claim or the original query.

Your MDP designs an active retrieval process, which can be considered as a special variant of active retrieval methods where we do not use the LLM to analyze between retrieval steps. So based on my idea, I cannot recognize this paper's contribution now.


[1] Asai A, Wu Z, Wang Y, et al. Self-rag: Learning to retrieve, generate, and critique through self-reflection[J]. 2024.

[2] Ugare S, Gumaste R, Suresh T, et al. IterGen: Iterative Semantic-aware Structured LLM Generation with Backtracking[J]. arXiv preprint arXiv:2410.07295, 2024.

**Questions:**

1. I would like to see comparison results to active RAG methods.


2. Please change figures to Vector graphics

---

### Official Review · Reviewer_D9zv · 2025-10-31

**Soundness:** 3
**Presentation:** 3
**Contribution:** 2
**Rating:** 4
**Confidence:** 3

**Summary:**

This paper introduces McsE, a heuristic search framework for evidence discovery that formulates the task as a Markov Decision Process and uses large language models (LLMs) as reward functions to estimate the strength of evidence supporting complex claims. The framework explicitly models both independent and collaborative (multi-hop) support among context sentences, enabling the discovery of reasoning chains rather than isolated evidence. Experiments on several datasets (HoVer, PubMedQA, CovidET, MultiRC) show that McsE outperforms strong statistical, embedding-based, and LLM-based baselines, particularly for multi-hop evidence extraction. Ablation studies and analyses are provided to support the value of the main components.

**Strengths:**

- This paper addresses an important and challenging problem: discovering collaborative, multi-hop evidence for claims in a general, task-agnostic way.
- The authors provide a Novel reformulation of evidence discovery as a Markov Decision Process with LLM-based reward functions, explicitly modeling both independent and collaborative support.
- Evaluation across multiple datasets and domains, with consistent improvements over strong baselines, with ablation studies and analysis of prompt design and collaborative reward modeling.

**Weaknesses:**

- High computational cost limits scalability and real-world applicability, especially for large candidate sets or longer reasoning chains.
- Relies on LLMs for reward estimation, risking propagation of LLM hallucinations or biases as 'evidence'.
- Evaluation on some datasets may be biased by LLM-generated ground truth and limited manual annotation.

**Questions:**

- Can the authors provide more details on the manual annotation/correction process for datasets with GPT-4-generated evidence? How is annotation quality and inter-annotator agreement ensured?
- How robust is the LLM-based reward signal to model drift, prompt variations, or out-of-domain contexts? Have you tested the framework with other LLMs or under distribution shift?

---

### Official Review · Reviewer_x8v5 · 2025-11-01

**Soundness:** 3
**Presentation:** 2
**Contribution:** 2
**Rating:** 2
**Confidence:** 4

**Summary:**

This paper proposes Markov-chain style Evidence (McsE), a framework for evidence discovery that retrieves supporting evidence for input claims using markov chain search. The authors argue that current methods fail to differentiate relevance and support, and these approaches struggle to identify collaborative / inter-sentence interactions, yet oftentimes multiple sentences must be combined to validate a complex claim. Therefore, McsE formulates evidence discovery as a multi-step search process with MDP, in which the search process leverages LLM as a scoring function to estimate evidence strength. McsE combines independent and collabirative supports, and also considers cumulative rewards in the value funtion to guide the iterative selection of evidence. To validate the effectiveness of the proposed method, the authors perform extensive experiments to show that McsE can outperform existing retriever / ranker baselines.

**Strengths:**

1. The authros identify critical challenges in evidence discovery / retrieval and motiviated by these challenges, they proposed McsE to leverage LLMs to guide the evidence discovery process with MDP search.

2. The authors introduce a novel method for evaluating evidence strength by using LLMs to score evidence for input claims, and thus constructing a robust value function to guide to search of the evidence chain.

3. McsE outperforms serveral baselines across three diverse benchmarks, demonstrating advantages in retriving and evaluating evidence for complex, multi-hop claims.

**Weaknesses:**

1. There's limited novelty within the proposed evidence discover method, as there are already a few papers with MCTS- or MDP-drive process for search / retrieval. Similar works like MCTS-RAG (https://arxiv.org/pdf/2503.20757), ETS (https://aclanthology.org/2025.acl-long.1175.pdf) and other chain of evidence / search should also be discussed and potentially included as baselines.

2. Lack of direct downstream performance (e.g., RAG /fact verification accuracy etc.). In addition, the entire framework's effectiveness relies on the LLM's ability to score with "according to" prompt; the authors should provide more analysis on prompt / scoring (log-p, verbalized probs etc.) selection and their effectiveness of the overall perofrmance.

3. The McsE framework is significantly more expensive than traditional retriever / ranker models, as it involves a complex heuristic search and multiple calls to an LLM to estimate value. Such computation costs are not disscussed in details.

**Questions:**

See weakness.

Also, there are some mismatches on figure / table captions, for example line 244 - 245 mentions figure 2, but should actually be figure 3.

---

### Meta-Review · Area_Chair_yS4G · 2025-12-28

**Summary:**

This paper propose a framework for evidence discovery for question answering and multi-hop reasoning tasks.  The goal is to find "Markov-chain-style evidence", which considers is evidence that both independently supports a claim as well as collaboratively supports it in the context of evidence already retrieved. This method is evaluated across three domains and compared to single-step retrieval methods, but not past multi-step retrieval methods.

Strengths

- Interesting problem space and problem formulation

- Evaluation covers three benchmarks and the proposed method outperforms baselines

Weaknesses

- Limited novelty: x8v5 and xZfh both point out relevant prior work to discuss/compare to. There is an extensive prior literature on methods for iterative retrieval. This paper needs to compare to those conceptually as well as empirically. This is a critical weakness.

- High computational costs: D9zv points out high computational costs for the search process given the use of LLMs internal to search.  Although this is potentially not a critical weakness, the paper should measure this, compare to baselines that are compute-matched as much as possible, and take a stand on whether the performance improvements are worth the higher computational cost.

**Reviewer Concerns:**

x8v5 and xZfh both point out relevant prior work to discuss/compare to.

D9zv points out high computational costs for the search process given the use of LLMs internal to search.

No rebuttal was given

**Reviewer Scores:**

No rebuttal was given.

---

### Decision · Program_Chairs · 2026-01-26

Reject